# SCF^FBXW11^ Complex Targets Interleukin-17 Receptor A for Ubiquitin–Proteasome-Mediated Degradation

**DOI:** 10.3390/biomedicines12040755

**Published:** 2024-03-28

**Authors:** Ben Jin, Sayed Ala Moududee, Dongxia Ge, Pengbo Zhou, Alun R. Wang, Yao-Zhong Liu, Zongbing You

**Affiliations:** 1Southeast Louisiana Veterans Health Care System, New Orleans, LA 70112, USA; bjin2@tulane.edu (B.J.); smoududee@tulane.edu (S.A.M.); 2Department of Structural & Cellular Biology, Tulane University, New Orleans, LA 70112, USA; 3Department of Orthopaedic Surgery, Tulane University, New Orleans, LA 70112, USA; dge@tulane.edu; 4Department of Pathology and Laboratory Medicine, Weill Cornell Medicine, New York, NY 10065, USA; pez2001@med.cornell.edu; 5Department of Pathology and Laboratory Medicine, Tulane University, New Orleans, LA 70112, USA; awang2@tulane.edu; 6Department of Biostatistics and Data Science, Tulane University, New Orleans, LA 70112, USA; yliu8@tulane.edu; 7Tulane Cancer Center and Louisiana Cancer Research Consortium, Tulane University, New Orleans, LA 70112, USA; 8Tulane Center for Stem Cell Research and Regenerative Medicine, Tulane University, New Orleans, LA 70112, USA; 9Tulane Center for Aging, Tulane University, New Orleans, LA 70112, USA

**Keywords:** FBXW11, IL-17RA, ubiquitin–proteasome-mediated degradation

## Abstract

Interleukin-17 (IL-17) is a pro-inflammatory cytokine that participates in innate and adaptive immune responses and plays an important role in host defense, autoimmune diseases, tissue regeneration, metabolic regulation, and tumor progression. Post-translational modifications (PTMs) are crucial for protein function, stability, cellular localization, cellular transduction, and cell death. However, PTMs of IL-17 receptor A (IL-17RA) have not been investigated. Here, we show that human IL-17RA was targeted by F-box and WD repeat domain-containing 11 (FBXW11) for ubiquitination, followed by proteasome-mediated degradation. We used bioinformatics tools and biochemical techniques to determine that FBXW11 ubiquitinated IL-17RA through a lysine 27-linked polyubiquitin chain, targeting IL-17RA for proteasomal degradation. Domain 665-804 of IL-17RA was critical for interaction with FBXW11 and subsequent ubiquitination. Our study demonstrates that FBXW11 regulates IL-17 signaling pathways at the IL-17RA level.

## 1. Introduction

Cytokines of the interleukin-17 (IL-17) family play various functions in innate and adaptive immune responses [1]. The IL-17 family consists of six members: IL-17A, IL-17B, IL-17C, IL-17D, IL-17E, and IL-17F, and there are five canonical IL-17 receptors (IL-17Rs) that interact with them, including IL-17RA, IL-17RB, IL-17RC, IL-17RD, and IL-17RE [1]. Canonical IL-17Rs are unique because of their two conserved structural features: two extracellular fibronectin II-like domains and one intracellular domain named similar expression to fibroblast growth factor genes and IL-17R (SEFIR) [2]. In addition to the canonical IL-17Rs, an uncanonical receptor, CD93, was reported to be specifically expressed on the surface of group 3 innate lymphoid (ILC3) cells [3]. The ligand and receptor association between IL-17 cytokines and IL-17Rs is complex due to various binding forms. Among them, the most well-studied IL-17 cytokines are IL-17A and IL-17F, which share the highest structural conservation among the IL-17 cytokines [1]. The IL-17A homodimer, the IL-17F homodimer, and the IL-17A/F heterodimer interact with the IL-17RA/RC heterodimer or the IL-17RC homodimer, inducing expression of various downstream genes [4]. IL-17RA heterodimerizes with IL-17RD, binding with the IL-17A homodimer [5]. The IL-17RA/RB heterodimer binds with the IL-17E homodimer [6,7]. IL-17RA also associates with IL-17RE to bind with the IL-17C homodimer [8]. IL-17D was reported to bind CD93, regulating colonic inflammation [3]. Although IL-17B was originally considered as the ligand for IL-17RB, it is believed to be a competitor against IL-17E in binding to IL-17RB [9]. So far, IL-17RA is best known as the common dimerization partner with all other canonical IL-17Rs, such as IL-17RC, IL-17RB, IL-17RD, and IL-17RE, indicating its unique functions in IL-17 signaling. Previous studies have delineated the special intracellular structures of IL-17RA beyond the SEFIR domain that are different from those of other canonical IL-17Rs [10,11].

Ubiquitylation is one of the post-translational modifications (PTMs) of proteins, achieving quality control of cellular responses [12]. There are three main steps of ubiquitylation. Ubiquitin is first activated by adenosine triphosphate (ATP) and the E1 enzyme. Then, the activated ubiquitin is conjugated to the E2 enzyme. The last step is the addition of a ubiquitin or polyubiquitin chain to a substrate by a particular E3 ligase [13]. There are more than 600 E3 ligases encoded by the human genome, and they usually form a functional complex with other components [14]. Based on how ubiquitin is transferred to substrates and the core domain of the complex components that facilitates this process, E3 ligases are further classified into two major groups: homologous to E6-AP C terminus (HECT) domain-containing E3 ligases [15] (accounting for around 5% of E3 ligases [16]) and really interesting new genes (RING) domain-containing E3 ligases [17] (accounting for around 95% of E3 ligases [16]). RING domain-containing E3 ligases are further classified into RING domain variants, individual E3 ligases, anaphase-promoting complex or cyclosome (APC/C) E3 ligases, and Cullin-Ring E3 ligases (CRLs) which contain a scaffold protein, Cullin [18]. The CRL E3 ligase family comprises more than 200 documented members and is the largest class of the E3 ligase family [19]. Based on different Cullins, the CRL E3 ligase complex is further classified into five subfamilies (Cullin1, Cullin2/5, Cullin3, Cullin4A/4B, and Cullin7). Among them, the best known are the Skp1-Cullin1-F box protein (SCF) E3 ligases [16]. In the case of CRL E3 ligases, before facilitating the transfer of ubiquitin to a substrate, they need to be activated by neddylation, where neural-precursor-cell-expressed, developmentally down-regulated gene 8 (NEDD8) interacts with Cullin [20]. After ubiquitylation, the outcome of the substrate is mostly dependent on which lysine is used for ubiquitylation. For lysine 48-linked polyubiquitylation, the substrate is usually subjected to proteasome-mediated degradation into small peptides. But for lysine 63-linked polyubiquitylation, the substrate is often activated to regulate signaling transduction and DNA repair [21]. The outcomes of the substrates vary when ubiquitin chains are formed using other lysine residues (K6, K11, K27, K29, and K33) or the first methionine residue (M1) [21].

The cascade of the IL-17 signaling pathway initiates from the binding of IL-17A/IL-17F with IL-17RA/IL-17RC. After dimerization IL-17RA and IL-17RC recruit NF-κB activator 1 (Act1) through the SEFIR domain, activated Act1 further recruits and ubiquitylates tumor necrosis factor receptor associated factor 6 (TRAF6) by K63-linked polyubiquitin chain. Polyubiquitylated TRAF6 then activates transforming growth factor-beta-activated kinase 1 (TAK1), leading to activation of the nuclear factor κB (NF-κB) and mitogen-activated protein kinase (MAPK) signaling pathways. Thereafter, IL-17-downsteam target genes, such as cytokines (tumor necrosis factor α, interleukin-1, and interleukin-6), chemokines (C-X-C motif ligand 1, C-X-C motif ligand 2, and C-C motif ligand 20), and matrix metalloproteinases (*MMP3*, *MMP7*, and *MMP9*), start to be transcribed. On the other hand, to restrain IL-17 signaling, deubiquitinase A20 is up-regulated upon IL-17 stimulation. Recruitment of A20 to IL-17RA through the C/EBPβ activation domain (CBAD) performs a negative feedback to deubiquitylate TRAF6, restricting IL-17-dependent activation of NF-κB and MAPK signaling [22]. In addition, Act1 is ubiquitylated by beta-transducin repeat-containing E3 ubiquitin protein ligase (bTrCP) through a K48-linked polyubiquitin chain for proteasomal degradation after prolonged stimulation of IL-17 [23]. Phosphorylation of Act1 by TANK binding kinase 1 (TBK1) also suppresses IL-17-mediated activation of NF-κB in a TRAF6-dependent fashion [24].

Our previous study reported that IL-17RA is constitutively phosphorylated by glycogen synthase kinase 3 (GSK3) at threonine 780, leading to ubiquitylation and proteasomal degradation [25]. However, the specific E3 ligase mediating this process is still elusive. In the current study, we found that IL-17RA is ubiquitylated by the SCF^FBXW11^ complex through K27-linked polyubiquitin. An intracellular 665-804 domain of IL-17RA is critical for ubiquitylation and degradation. Taking this together with previous reports [23,26], the SCF^FBXW11^ complex regulates the IL-17 signaling pathway not only at the IL-17RA level, but also at the Act1 and IκBα levels.

## 2. Materials and Methods

**Mammalian cell culture.** Human prostate cancer cell lines (22Rv1, PC-3, and LNCaP), human normal keratinocytes (HaCaTs), a human embryonic kidney cell line (HEK293T), a human cervical cancer cell line (HeLa), a human lung cancer cell line (A549), and a human monocytic leukemia cell line (THP-1) were purchased from the American Type Culture Collection (ATCC, Manassas, VA, USA). Human skin squamous cell carcinoma cell line A-431 was kindly gifted by Dr. Shitao Li of Tulane University. The human endometrial cancer cell line Ishikawa was kindly gifted by Dr. Matthew Burow of Tulane University. Human colon cancer cell lines HCT116 and DLD1 were kindly gifted by Dr. Lin Zhang at the University of Pittsburgh. A stable cell line (HEK293-IL-17RA) overexpressing exogenous Flag-IL-17RA was established in Dr. You’s lab [25]. A549 FBXW11 WT and FBXW11 KO cell lines were kindly gifted by Dr. Friedemann Weber of the Institute for Virology, FB10-Veterinary Medicine, Justus-Liebig University, with the permission of Dr. Veit Hornung of Ludwig-Maximilians-Universität Munich [27] The HaCaT, HEK293T, HeLa, A549, A-431, Ishikawa, HCT116, DLD1, HEK293-IL17RA, A549 FBXW11 WT, and A549 FBXW11 KO cell lines were maintained in high-glucose Dulbecco’s Modified Eagle Medium (DMEM) (Genesee Scientific, San Diego, CA, USA, #25-500) supplied with 10% fetal bovine serum (FBS, ATCC, Manassas, VA, USA, #30-2022). The THP-1, 22Rv1, and LNCaP cell lines were maintained in RPMI-1640 (Genesee Scientific, San Diego, CA, USA, #25-506) medium supplied with 10% FBS. The PC-3 cell line was maintained in Kaighn’s Modification of Ham’s F-12 (F-12K) (ATCC, Manassas, VA, USA, #30-2004) supplied with 10% FBS. All cell lines were maintained in a humidified condition at 37 °C with 5% carbon dioxide. Where indicated, cells were treated with cycloheximide (CHX, Millipore Sigma, Saint Louis, MO, USA #01810), MG132 (Millipore Sigma, Saint Louis, MO, USA #M8699; Cayman chemical company, Ann Arbor, MI, USA, #10012628), bortezomib (Selleckchem, Houston, TX, USA, #S1013), MLN4924 (Selleckchem, Houston, TX, USA #S7109), and imidazole (Fisher scientific, Houston, TX, USA, #A10221.36) dissolved in dimethyl sulfoxide (DMSO) (Fisher scientific, Houston, TX, USA, #BP231-100).

**Plasmids.** The following plasmids were purchased from Addgene (Watertown, MA, USA): HA-Ubiquitin WT (#17608), Myc-FBXW7 ∆F box (#16652), Myc-FBXW5 (#19905), Myc-FBXW1A (#20718), and Myc-Skp2 (#19947). The following plasmids were kindly gifted by Dr. Michael Pagano of New York University: Flag-FBXW2, Flag-FBXW4, Flag-HA-FBXW8, Flag-HA-FBXW9, Flag-HA-FBXW11, and Flag-HA-FBXW12. The following plasmids were kindly gifted by Dr. Hua Lu of Tulane University: His-ubiquitin WT, His-ubiquitin K6R, His-ubiquitin K11R, His-ubiquitin K27R, His-ubiquitin K29R, His-ubiquitin K33R, His-ubiquitin K48R, and His-ubiquitin K63R [28]. Plasmids of lentiCRISPRv2 (addgene, #52961), pMD2.G (addgene, #12259) and psPAX2 (addgene, #12260) were also kindly gifted by Dr. Hua Lu of Tulane University. The Flag-IL-17RA full-length plasmid was obtained from Dr. Xiaoxia Li [29], and it was used as a template to create Flag-IL-17RA ∆729-773 and Flag-IL-17RA ∆665-804 truncation mutants. The Myc-His-IL-17RA plasmid was generated as previously described [30].

**CRSIPR/Cas9-mediated gene knock-out.** The exon sequence to be targeted (5′-GTGGACGACACAACTTGCAGAGG-3′) was selected by the CHOPCHOP online tool (http://chopchop.cbu.uib.no/, accessed on 15 November 2022), and this target was validated by Dr. Friedemann Weber and Dr. Veit Hornung [27]. Standard de-salted oligos (Oligo 1: 5′-CACCGGTGGACGACACAAC TTGCAG-3′; Oligo 2: 3′-CCACCTGCTGTGTTGAACGTCCAAA-5′) were synthesized by Eurofins Genomics (Framingham, MA, USA) and were diluted to 100 μM in sterile water. The GeCKO system was applied to generate a lentiviral CRISPR tool following well-established protocols from Dr. Feng Zhang’s lab [31,32]. Oligo 1 and Oligo 2 were annealed by T4 polynucleotide kinase (NEB, Ipswich, MA, USA, #M0201S) at 37 °C for 30 min and 95 °C for 5 min; then, the temperature was ramped down to 25 °C at 5 °C/min. The lentiCRISPRv2 was digested by BsmBI-v2 (NEB, Ipswich, MA, USA, #R0739S) at 55 °C for 15 min and heated to be inactivated at 80 °C for 20 min before chilling on ice. Digested lentiCRISPRv2 and annealed oligos were ligated with T4 DNA ligase (NEB, Ipswich, MA, USA,#M0202S) at 16 °C overnight and were heated to be inactivated at 65 °C for 10 min before chilling on ice. A quantity of 5 µL of ligation product was transformed into 50 µL of Stbl3 competent cells (Invitrogen, Carlsbad, CA, USA, #C737303). Insertion of guide RNA into the lentiviral CRISPR plasmid was verified by Sanger sequencing under the U6-forward primer (5′-GACTATCATATGCTTACCGT-3′) before packaging of the lentiviruses. HEK293T cells were placed into a 10 cm dish 24 h before transfection and supplied with 7 mL of Dulbecco’s Modified Eagle’s Medium (Genesee Scientific, San Diego, CA, USA, #25-500) containing 10% fetal bovine serum. When the confluency reached 70–80%, jetPRIME (Polyplus, New York, NY, USA, #101000046) was used to carry out co-transfection according to the manufacturer’s instructions. The amount of each plasmid was 5 μg recombinant lentiCRISPR, 2 μg pMD2.G, and 3 μg psPAX2. The plasmids were incubated for 12 h before replacing the culture medium with 6 mL of fresh complete culture medium. The supernatant containing lentiviruses was collected two times, at 48 h and 96 h post-transfection. The supernatant containing lentiviruses was centrifuged at 1000 rpm for 5 min at room temperature and aliquoted to be stored at −80 °C. Twenty-four hours before infection with lentiviruses containing guide RNA targeting FBXW11 or an empty vector, Ishikawa cells were seeded into a well of a 6-well plate. When the cell confluency reached 30–50%, a mixture of 500 μL of fresh culture medium, 500 μL of virus medium, and 8 µg/mL of polybrene was added to the cells. Twenty-four hours post the first infection, the infection was repeated once. Forty-eight hours post the second infection, 1 μg/uL puromycin was used to select cells with successful infection. Seven days later, the surviving cells were serially diluted and placed into 96-well plates at a density of 0.5 cell per well. Puromycin selection was maintained for another 3–4 weeks until colonies were visible by the naked eye. Colonies were picked up for amplification before isolating genomic DNA using a Quick-DNA miniprep Plus kit (Zymo research, Irvine, CA, USA, #D4068). Standard PCR with a forward primer (5′-TATCGGTGGTATGCTGTTTCTG-3′) and a reverse primer (5′-TCTCGTAGGCCAC TGATAATTT-3′) was applied to amplify the target DNA sequence. Sanger sequencing was used to determine the genotypes of clones with the PCR primers. One clone (termed C1) that contained a 4-base pair deletion at the target site (5′-GTGGACGACACAA----CAGAGG-3′) was selected as the FBXW11 knock-out cell line.

**Transfection of plasmids and small interference RNA (siRNA).** Human IL-17RA siRNA (#sc-40037) and nontargeting control RNA (#sc-37007) were obtained from Santa Cruz Biotechnology (Dallas, TX, USA). Human FBXW1A siRNAs (#D-003463-01-0002, D-003463-02-0002, D-003463-03-0002, and D-003463-04-0002) were purchased from Dharmacon (Lafayette, CO, USA). AllStars Negative Control RNA (#SI03650318) was obtained from QIAGEN (Germantown, MD, USA). Cells were transfected with plasmids or siRNA when the confluency reached around 70–80%. Transfection was performed using jetPRIME transfection reagent (Polyplus, New York, NY, USA, #114-15) according to the manufacturer’s instructions. After transfection, the cells were incubated for 12–24 h before replacing the culture medium with fresh culture medium, and subsequent treatments were conducted appropriately.

**Western blot analysis.** Cells were washed with phosphate-buffered saline (PBS) and lysed with radioimmunoprecipitation assay (RIPA) buffer (50 mM sodium fluoride, 0.5% NP-40, 10 mM sodium phosphate monobasic, 150 mM sodium chloride, 25 mM Tris (pH 8.0), 2 mM ethylenediaminetetraacetic (EDTA), and 0.2 mM sodium vanadate) with a fresh supplement of 1× protease inhibitor cocktail (PIC, Millipore Sigma, Saint Louis, MO, USA, #P8849). After incubation on ice for 15 min, the cell lysate was centrifuged at 13,000× *g* for 15 min at 4 °C. The supernatant of the whole cell lysate was transferred into a new microcentrifuge tube and boiled with 3× sample loading buffer (208.1 mM sodium dodecyl sulfate (SDS), 30% glycerol, 187.5 mM Tris (pH 6.8), 15% β-mercaptoethanol, and 14.9 mM bromophenol blue) at 100 °C for 5–10 min. Around 60–120 µg of total protein was subjected to sodium dodecyl sulfate polyacrylamide gel electrophoresis (SDS-PAGE) and transblotted to the polyvinylidene fluoride membranes (PVDF, Genesee Scientific, San Diego, CA, USA, #83-646R). The blots were blocked with 2.5% bovine serum albumin (Millipore Sigma, Saint Louis, MO, USA, #A3294) solution containing 0.02% sodium azide (Millipore Sigma, Saint Louis, MO, USA, #S2002). The blots were probed with the following primary antibodies at room temperature for 1–2 h or at 4 °C overnight: anti-GAPDH (Millipore Sigma, Saint Louis, MO, USA, #MAB374, 1:5000 dilution), anti-Flag M2 (Millipore Sigma, Saint Louis, MO, USA, #F3165, 1:20,000 dilution), anti-IL-17RA (Santa Cruz Biotechnology, Dallas, TX, USA, #sc-376374, 1:500 dilution), anti-c-Myc (Santa Cruz Biotechnology, Dallas, TX, USA, #sc-40, 1:1000 dilution), anti-c-Myc (Novus, Centennial, CO, USA, #NB600-335, 1:1000 dilution), anti-HA (Santa Cruz Biotechnology, Dallas, TX, USA, #sc-7392, 1:1000 dilution), anti-FBXW7 (Bethyl Laboratories, Montgomery, TX, USA, #A301-720A, 1:15,000 dilution), anti-p-Akt (Cell Signaling Technology, Danvers, MA, USA, #9271, 1:500 dilution), anti-Akt (Santa Cruz Biotechnology, Dallas, TX, USA, #sc-81434, 1:100 dilution), anti-p-IκBα (Cell Signaling Technology, Danvers, MA, USA, #2859, 1:1000 dilution), anti-IκBα (Cell Signaling Technology, Danvers, MA, USA, #4814, 1:1000 dilution), anti-p-JNK (Cell Signaling Technology, Danvers, MA, USA, #9255, 1:500 dilution), anti-JNK (Cell Signaling Technology, Danvers, MA, USA, #9252, 1:1000 dilution), anti-p-ERK1/2 (Santa Cruz Biotechnology, Dallas, TX, USA, #sc-7383, 1:500 dilution), anti-ERK1/2 (Cell Signaling Technology, Danvers, MA, USA, #4695, 1:1000 dilution), anti-p-p38 MAPK (Santa Cruz Biotechnology, Dallas, TX, USA, #sc-166182, 1:500 dilution), anti-p38 MAPK (Cell Signaling Technology, Danvers, MA, USA, #8690, 1:1000 dilution), anti-NF-κB p65 (Cell Signaling Technology, Danvers, MA, USA, #6956, 1:500 dilution), anti-β-tubulin (Cell Signaling Technology, Danvers, MA, USA, #2128, 1:1000 dilution), and anti-Histone H3 (Cell Signaling Technology, Danvers, MA, USA, #4499, 1:2000 dilution). Li-Cor IRDye680- and IRDye800-conjugated secondary antibodies (Lincoln, NE, USA) were incubated at room temperature for 45 min at a dilution of 1: 10,000 and 1: 5000, respectively. The blots were scanned with a Li-COR Odyssey 9120 Digital Imaging system (Lincoln, NE, USA). When necessary, the blots were stripped with a stripping buffer (62 mM Tris-HCl (pH 6.7), 2% SDS, and 100 mM β-mercaptoethanol) and re-scanned to confirm stripping efficiency before probing for another antigen of interest. Adobe illustrator 2024 (San Jose, CA, USA) was utilized to organize the images.

**Co-immunoprecipitation (co-IP).** HEK293T cells were seeded at a density of 1 × 10^6^ cells per 6 cm dish or 4.5 × 10^6^ cells per 10 cm dish, approximately 20 h prior to transfection. Quantities of 3 mL and 7 mL of DMEM (with 10% FBS) were added to the 6 cm and 10 cm dishes, respectively. Transient transfection of plasmids was conducted using jetPRIME (Polyplus, New York, NY, USA, #114) according to the manufacturer’s instructions. Forty-eight hours post-transfection, protein was extracted with an IP lysis buffer (50 mM Tris-HCl (pH 7.5), 0.5% Nonidet P-40, 1 mM EDTA, and 150 mM NaCl). IP lysis buffer was freshly supplemented with 1× PIC and 1 mM 1,4-dithiothreitol (DTT, Thermo Fisher, Lenexa, KS, USA, #R0861). A proper amount of lysis buffer was added and vortexed vigorously for 10 s, followed by incubation on ice for 30 min and vortexing every 10 min. Cell debris was removed by centrifugation at 13,000× *g* at 4 °C for 15 min. The supernatant was transferred into a new centrifuge tube, and the protein concentration was quantified with the Bradford assay [33]. Immunoprecipitation was carried out by mixing 500–800 µg of whole cell lysate and 1–2 µg of anti-Flag M2 antibody (Sigma Millipore, St. Louis, MO, USA, #F3165), 1–2 µg of anti-c-Myc antibody (Santa Cruz Biotechnology, Dallas, TX, USA, #sc-40), or 12 µg of anti-FBXW7 antibody (Bethyl Laboratories, Montgomery, TX, USA, #A301-720A). Then, 1–2 h later, 20 µL of 75% (*v*/*v*) rProtein A agarose resin beads (Genesee Scientific, San Diego, CA, USA, #20-525) was added to each sample and incubation was maintained with gentle rocking at 4 °C overnight. The beads were washed with 1 mL of pre-chilled IP lysis buffer (freshly supplemented with 1 mM DTT and 1× PIC) 3 times. The precipitated proteins were eluted in 20 µL of 2× sample loading buffer with boiling at 100 °C for 5 min. The samples were subjected to Western blot analysis.

**Ubiquitylation assay.** Forty-two hours post-transfection, the cells were treated with 20 µM MG132 for 6–8 h. The cells were collected and washed with PBS once. When Ni-NTA resins (Thermo Fisher, Lenexa, KS, USA, #88221) were used, the cell pellet was aliquoted into two parts: 25% of the cells were lysed with RIPA buffer, freshly supplemented with 1× PIC, on ice for 10 min, and 75% of the cells were lysed with fresh Buffer B (8 M Urea, 0.1 M Na_2_HPO_4_/NaH_2_PO_4_ (pH 8.0), 0.01 M Tris-Cl (pH 8.0), 10 mM β-mecaptoethanol, and 25 mM imidazole) at room temperature for 10 min. The supernatants were collected after centrifugation at 13,000× *g* for 15 min at 4 °C and room temperature, respectively. The whole cell lysate extracted with RIPA buffer was subjected to SDS-PAGE, and the signal intensity of IL-17RA detected with Western blotting was used to adjust the amount of cell lysate of each sample applied to subsequent pull-down assays in order to ensure that equal amounts of proteins were used for each pull-down reaction. A quantity of 30 µL of 50% Ni-NTA resins was washed three times with Buffer B and then resuspended in 100 µL of Buffer B. The resins were incubated with the proper volume of cell lysate extracted with Buffer B at room temperature for 4 h. The Ni-NTA resins were washed with 1 mL of Buffer B 3 times and 1 mL of fresh Buffer C (8 M Urea, 0.1 M Na_2_HPO_4_/NaH_2_PO_4_ (pH 6.3), 0.01 M Tris-Cl (pH 6.3), and 10 mM β-mecaptoethanol) 2 times. The samples were processed for elution by being boiled twice in 1× sample loading buffer containing 300 mM imidazole at 100 °C for 3 min each time. When rProtein A agarose resins were used, the cell pellet was boiled in 120 µL of denaturing IP lysis buffer (50 mM Tris-HCl (pH 7.5), 0.5% Nonidet P-40, 1 mM EDTA, 150 mM NaCl, freshly supplemented with 1× PIC, 1 mM DTT, and 1% SDS) at 100 °C for 10 min. After centrifugation at 13,000× *g* at room temperature for 15 min, 110 µL of supernatant was collected. A quantity of 20 µL of supernatant was aliquoted into a new tube as a whole cell lysate sample and boiled with 3× sample loading buffer for 5 min. The rest of the 90 µL supernatant was diluted with freshly prepared IP lysis buffer (without SDS) until the concentration of SDS was less than 0.1%. Before immunoprecipitation, a direct Western blot analysis was conducted to decide the proper amount of cell lysate needed. During immunoprecipitation, 3 µg anti-HA antibody (Santa Cruz Biotechnology, Dallas, TX, USA, #sc-7392) was used to pull down ubiquitin-conjugated IL-17RA, which was incubated at 4 °C with gentle agitation for 1–2 h. Then, 20 µL of 75% (*v*/*v*) rProtein A agarose resins (Genesee Scientific, San Diego, CA, USA, #20-525) were added to each sample. Samples were incubated with gentle rocking at 4 °C overnight. The resin beads were washed with 1 mL pre-chilled IP lysis buffer (freshly supplemented with 1 mM DTT and 1× PIC) 3 times and boiled with 20 µL 2× loading buffer at 100 °C to elute the precipitated proteins, which was repeated once to achieve a thorough elution. The samples were subjected to subsequent Western blot analysis.

**Nuclear and cytoplasmic protein extraction.** For both the A549 and Ishikawa cell lines, 24 h before treatment, 2.0 × 10^6^ FBXW11 WT and 2.5 × 10^6^ FBXW11 KO cells were seeded into 10 cm dishes. A quantity of 20 ng/mL recombinant human IL-17A (rhIL-17A, R&D Systems, Minneapolis, MN, USA, #7955-IL-025/CF) was used to treat the cells for 10 or 30 min. According to the manufacturer’s instructions, cytoplasmic and nuclear proteins were extracted using an NE-PER nuclear and cytoplasmic extraction kit (Thermo Scientific, Lenexa, KS, USA, #78833). In brief, FBXW11 WT and FBXW11 KO cells were collected with trypsin-EDTA. After centrifugation at 500× *g* for 5 min at 4 °C, the supernatant was discarded. Then, 200 μL ice-cold cell extraction reagent I was added and vortexed vigorously for 15 s to lyse the cell membranes, followed by incubation for 10 min and the addition of 11 μL ice-cold cell extraction reagent II. The lysates were vortexed for 5 min and incubated on ice for 1 min. The supernatant was transferred into a new tube after centrifugation at 13,000× *g* for 15 min. The insoluble pellet was resuspended with 50–100 μL nuclear extraction reagent and vortexed for 15 s, followed by incubation on ice for 40 min with vortexing every 10 min. The supernatant was collected after centrifugation at 13,000× *g* for 15 min. The protein concentration was quantified with a Take3 microplate spectrometer (Agilent BioTeck, Synergy H1, Santa Clara, CA, USA).

**Quantitative real-time PCR (qPCR) analysis.** The FBXW11 WT and FBXW11 KO cell lines (A549 and Ishikawa) were treated with 20 ng/mL rhIL-17A (R&D Systems, Minneapolis, MN, USA, #7955-IL-025/CF), rhIL-17B (R&D Systems, Minneapolis, MN, USA, #8129-IL-025/CF), rhIL-17C (R&D Systems, Minneapolis, MN, USA, #9640-IL-025/CF), rhIL-17E (R&D Systems, Minneapolis, MN, USA, #8134-IL-025/CF), or rhIL-17F (R&D Systems, Minneapolis, MN, USA, #1335-INS-025/CF) for 2 h. According to the manufacturer’s instructions, total RNA was isolated using Trizol reagent (Ambion, Carlsbad, CA, USA, #15596018) and quantified using a Take3 microplate spectrometer (Agilent BioTeck, Synergy H1, Santa Clara, CA, USA). A quantity of 1000 ng of RNA was used to synthesize cDNA by reverse transcriptase (Takara, San Jose, CA, USA, #RR037A). The cDNA was diluted with DNase/RNase-free water 10-fold before performing qPCR. In each qPCR reaction, a mixture in one well of 384-well plates consisted of 2 μL PCR primer mix (with a final concentration of 1 μM and the primer sequences shown in Table 1), 4 μL SYBR Green Master mix (Applied Biosystems, Vilnius, Lithuania, #A25742), 0.8 μL diluted cDNA, and 1.2 μL DNase/RNase-free water. The following default PCR program was conducted: stage 1: 95 °C 5 min; stage 2: 95 °C 15 s, 60 °C 1 min, ×40 cycles; melt curve: 95 °C 15 s, 60 °C 1 min, 95 °C 15 s. The ∆∆Ct method [34] was used to calculate the fold change of gene expression. GAPDH was used as a loading control, and the WT-Control group was used for calibration. Fold change relative to the control group of FBXW11 WT for each cell line was calculated as 2−∆∆Ct [(∆∆Ct = sample (CtTarget − CtGAPDH) − calibrator (CtTarget − CtGAPDH)].

**Bioinformatics analysis.** The Human Protein Atlas database was analyzed to figure out IL-17RA mRNA levels across multiple human cell lines (https://www.proteinatlas.org/; accessed on 27 November 2022) [35]. Putative phosphodegrons of IL-17RA were predicted by Dr. Pengbo Zhou, and the Eukaryotic Linear Motif (ELM) resource was explored to further validate these predictions [36]. Multiple sequence alignment of the tryptophan–aspartic acid (WD) repeat domain of FBXW family members was conducted with CLUSTAL Omega O (1.2.4) (https://www.uniprot.org/align, accessed on 12 December 2021) [37]. Proteomics data of glioblastoma multiforme and uterus corpus endometrial carcinoma (UCEC) were downloaded from the Clinical Proteomic Tumor Analysis Consortium (CPTAC) (https://pdc.cancer.gov/pdc/, accessed on 1 June 2022) [38,39] and the UALCAN database (https://ualcan.path.uab.edu/, accessed on 1 December 2022) [40]. Z-values of IL-17RA and FBXW11 protein levels in glioblastoma multiforme, UCEC, and corresponding normal brain and endometrial (with or without enrichment)/myometrial tissues were provided by Dr. Darshan Shimoga Chandrashekar and Dr. Sooryanarayana Varambally of the University of Alabama at Birmingham, which were used to carry out Pearson’s correlation analysis. Phosphorylation abundance of human IL-17RA in UCEC was analyzed using the LinkedOmicsKB platform (https://kb.linkedomics.org/#, accessed on 1 June 2023) [41].

**Statistical analysis.** All in vitro experiments were repeated at least 3 times unless it is indicated otherwise in the figure legends. GraphPad Prism software (Version 9.5.1, 733) was applied to carry out all statistical analyses. A two-way analysis of variance (ANOVA) was conducted to determine the source of variation and its significance when the cells were treated with cycloheximide. Simple linear regression was used to calculate the half-life times of proteins when the cells were treated with cycloheximide. One-way ANOVA or the unpaired Student’s *t*-test were applied to determine statistical significance for quantitative data. A two-tailed *p*-value < 0.05 was considered statistically significant.

## 3. Results

### 3.1. Inhibition of Proteasome and Cullin-Ring Ligase (CRL) Complex Increases IL-17RA Protein Stability

Our previous study reported that exogenous IL-17RA was degraded through a ubiquitin–proteasome system (UPS) [25]. But a specific E3 ligase mediating this process has not been identified. The CRL E3 ligase family contains the most diverse subunits with more than 200 E3 ligase members [16]. One of the core subunits of the CRL E3 ligase complex is a scaffold protein, Cullin. Neddylation of Cullin by interaction with NEDD8 activates the E3 ligase complex to ubiquitylate the substrates [42]. To verify if IL-17RA is degraded through the proteasome and particularly by the CRL complex, full-length human IL-17RA was transiently overexpressed in HEK293T cells and then the cells were treated with cycloheximide (CHX). When protein synthesis was inhibited by CHX, the total IL-17RA protein level decreased rapidly. A proteasome inhibitor, MG132, and a neddylation inhibitor, MLN4924, delayed degradation of exogenous IL-17RA (Figure 1A). Using the Human Protein Atlas (HPA) database, we examined mRNA levels of IL-17RA in different human cell lines and found that IL-17RA was ubiquitously expressed in human cell lines from different organs, although the expression levels were tremendously variable (Appendix A). Because multiple commercially available anti-IL-17RA antibodies did not work well to detect endogenous IL-17RA, we used small interference RNA to knock down IL-17RA in HaCaT and 22Rv1 cell lines and probed them with a G9 clone of a monoclonal antibody obtained from Santa Cruz Biotechnology (#sc-376374, Dallas, TX, USA). The results of Western blot analysis indicated that this G9 clone was able to specifically recognize endogenous IL-17RA (Appendix A). Therefore, we used this antibody in our subsequent experiments to detect endogenous and exogenous IL-17RA.

Endogenous IL-17RA levels in 22Rv1 cells decreased rapidly after CHX treatment, while MG132 and MLN4924 inhibited this reduction (Figure 1B). This result was consistent with that for exogenous IL-17RA in HEK293T cells (Figure 1A). Treatment with MG132 also stabilized endogenous IL-17RA in HaCaT cells (Figure 1C). These findings indicate that degradation of exogenous and endogenous IL-17RA was mediated by a ubiquitin–proteasome system, particularly by the CRL complex, as the neddylation inhibitor delayed the degradation. We conducted further studies on the impact of the neddylation inhibitor on endogenous IL-17RA levels in HaCaT, 22Rv1, and PC-3 cell lines using different dosages of MLN4924 over different time periods. As shown in Figure 1D–F, treatment with MLN4924 resulted in accumulation of endogenous IL-17RA and c-Myc, a well-known substrate ubiquitylated by a CRL E3 ligase F-box and WD repeat domain-containing 7 (FBXW7) [43].

Bortezomib is another proteasome inhibitor that has been approved by the US Food and Drug Administration (FDA) to treat multiple myeloma, mantle cell lymphoma, and acute allograft rejection [44]. HaCaT, THP-1, 22Rv1, and PC-3 cells were treated with different doses of bortezomib for 14 h. The results showed that endogenous IL-17RA levels were slightly accumulated (Appendix A). Combined treatment of CHX and bortezomib also showed that bortezomib slightly decreased degradation of endogenous IL-17RA (Appendix A). It has been reported that bortezomib at a nanomolar concentration only inhibited ~70% of proteasome activity, while MG132 at a micromolar concentration inhibited ~95% of proteasome activity [45], which explains why bortezomib only slightly prevented degradation of endogenous IL-17RA while MG132 dramatically prevented IL-17RA degradation. Besides proteasome-mediated degradation of damaged or misfolded proteins, autolysosomes also play a role in breaking down proteins through lysosomal enzymes [46]. To test if IL-17RA is also degraded through lysosomes, we treated HaCaT cells with different doses of the lysosome inhibitor imidazole for different time periods. The results showed that alteration of IL-17RA levels was not obvious, indicating that degradation of IL-17RA was probably not mediated through the lysosomes (Appendix A).

### 3.2. F-Box and WD Repeat Domain-Containing (FBXW) Proteins Are Predicted to Recognize IL-17RA

Having shown that IL-17RA was degraded by the ubiquitin–proteasome system, particularly through the CRL complex, we investigated which E3 ligase targets IL-17RA for ubiquitylation and degradation. To narrow down our targets from more than 600 E3 ligases encoded by the human genome [14], we first used bioinformatics tools to predict the potential candidates. We previously found that GSK3 phosphorylates IL-17RA at T780, leading to ubiquitylation and degradation of IL-17RA [25]. Therefore, we speculated that IL-17RA might have some phosphodegrons (short motifs with specific sequence patterns) that could be recognized by E3 ligases for ubiquitylation and degradation [16,47]. In the protein sequence of human IL-17RA, there are two candidate phosphodegrons that can be recognized by FBXW7 and FBXW1A/FBXW11, which are conserved across different species (Appendix A). FBXW proteins belong to the CRL E3 ligases. Because Cullin1 and S-phase kinase-associated protein 1 (Skp1) are required to form an E3 ligase complex, these CRL E3 ligases are called Skp1-Cullin1-F-box (SCF) E3 ligases [48]. In total, there are 10 members in the FBXW family, named FBXW1A through to FBXW12 (FBXW3 is the pseudogene of FBXW4 and FBXW6 is identical to FBXW8) [49]. The F-box domain and the WD repeat domain are conserved across different FBXW E3 ligases (Appendix A). Skp1 is an adaptor that bridges the F-box domain of the FBXW E3 ligase and Cullin1 to form a complex. The WD repeat domain is used to recognize a specific substrate for ubiquitylation [50]. Draberova et al. applied mass spectrometry to analyze components in the precipitates pulled down by recombinant Strep-Flag-tagged mouse IL-17A (SF-IL-17A). In addition to the well-known components of the IL-17 signaling pathway that interact with IL-17, such as IL-17RA, IL-17RC, Act1, and TRAF6, they also found Cullin1 and beta-TrCP1/2 (also known as FBXW1A/11) [51], indicating that FBXW1A and FBXW11 may physically bind to IL-17RA.

### 3.3. Several SCF E3 Ligases Are Associated with IL-17RA

Because FBXW family members utilize the conserved WD domain to recognize substrates, we used co-immunoprecipitation (co-IP) assays to screen the FBXW proteins that might bind with IL-17RA. In HEK293T cells, Flag-tagged IL-17RA was co-transfected with the Myc-tagged FBXW7 ∆F box, FBXW5, FBXW1A, and Skp2. Our reciprocal co-IP results showed that the FBXW7 ∆F box and FBXW5 had a strong association with IL-17RA, while FBXW1A had less binding with IL-17RA (Figure 2A). We also co-transfected Myc-His-tagged IL-17RA together with Flag-tagged FBXW2, FBXW4, FBXW8, FBXW9, FBXW11, and FBXW12. The co-IP results showed that the strongest binding partners of IL-17RA were FBXW9 and FBXW11 (Figure 2B). Because we used FBXW7 with truncation of the F-box, we then used HA-tagged full-length FBXW7 to examine the physical association, and the results showed that FBXW7 weakly bound to IL-17RA (Appendix A). Our co-IP data indicated that there are several E3 ligase candidates for IL-17RA, including FBXW1A, FBXW5, FBXW7, FBXW9, and FBXW11.

### 3.4. FBXW11 Ubiquitylates IL-17RA via K27-Linked Polyubiquitin in a Dose-Dependent Way

To determine which E3 ligase candidate has the highest activity to ubiquitylate IL-17RA, Flag-tagged IL-17RA, HA-tagged wild-type (WT) ubiquitin, and candidate E3 ligases were co-transfected into HEK293T cells, and MG132 was used to treat the cells before extracting whole cell lysates. Anti-HA antibody was used to pull down ubiquitylated substrates, and anti-Flag and anti-IL-17RA antibodies were used to probe ubiquitylated IL-17RA. The ubiquitylation assays showed that FBXW11 had the highest activity to ubiquitylate IL-17RA, while FBXW9 and FBXW1A had much lower activities than FBXW11 (Figure 3A). We also used Ni-NTA beads to pull down His-tagged ubiquitin and confirmed that FBXW11 had the highest activity to ubiquitylate IL-17RA (Appendix A). Ubiquitylation of endogenous IL-17RA by FBXW11 in THP-1 and HCT116 cell lines was dose-dependent (Figure 3B,C). Engineered ubiquitin variants can selectively bind to ubiquitin-binding domains and block recognition of natural ubiquitylation substrates; thus, they can be used to inhibit specific E3 ligase activity [52]. Ubv.Fw11.2 is such a ubiquitin variant that selectively prevents the formation of the SCF^FBXW11^ complex [53]. We applied Ubv.Fw11.2 in our ubiquitylation assays and found that Ubv.Fw11.2 slightly inhibited ubiquitylation of endogenous IL-17RA in THP-1 cells but not in HCT116 cells (Figure 3B,C). A possible reason for this difference may be due to the low basal level of ubiquitylation present in these two cell lines.

It is known that seven lysine (K) residues and the first methionine (M) residue in ubiquitin participate in forming ubiquitin chains. Of note, K11, K27, and K48 are involved in the proteasomal degradation of substrates [21]. In THP-1 and HCT116 cells, we co-transfected Flag-HA-FBXW11 along with His-tagged WT ubiquitin and a series of ubiquitin mutants with a single lysine-to-arginine substitution. The ubiquitylation assay results revealed that mutation of K27 into arginine (K27R) remarkably decreased ubiquitylation levels of IL-17RA in both cell lines, although mutation of other lysine residues also showed various degrees of reduction in ubiquitylation (Figure 3D,E). We also found that mutations of K48 and K63 slightly decreased ubiquitylation of IL-17RA in HCT116 cells (Figure 3E), which agreed with our previous report [25].

### 3.5. Overexpression of FBXW11 Accelerates Degradation of IL-17RA, While Knock-Out of FBXW11 Increases Protein Stability of IL-17RA

Having determined that FBXW11 is the E3 ligase involved in ubiquitylation of IL-17RA, we assessed the effects of FBXW11 on IL-17RA protein stability. Our lab previously established an HEK293 cell line named HEK293-IL-17RA that stably overexpresses Flag-IL-17RA [25]. We transfected Flag-FBXW4, Myc-FBXW5, Flag-HA-FBXW9, Myc-FBXW1A, or Flag-HA-FBXW11 into HEK293-IL-17RA cells and used CHX to inhibit protein synthesis. Western blot analysis showed that FBXW9 and FBXW11 dramatically accelerated degradation of Flag-IL-17RA (Figure 4A) compared to a vector control and other E3 ligases. In THP-1 cells, ectopic overexpression of Myc-FBXW1A, Flag-HA-FBXW11, and Flag-HA-FBXW9 enhanced degradation of endogenous IL-17RA. In the Ishikawa cell line, only FBXW11 remarkably accelerated degradation of endogenous IL-17RA (Figure 4B). These findings suggest that overexpression of FBXW11 decreased the protein stability of IL-17RA. Next, we examined the effects of FBXW11 knock-out (KO) on IL-17RA protein stability. The A549 FBXW11 KO cell line was generated using a clustered regularly interspaced short palindromic repeats (CRISPR)/CRISPR-associated protein 9 (Cas9) technique [27]. FBXW11 knock-out increased the basal levels of IL-17RA protein compared to the parental A549 FBXW11 WT cells. CHX treatment led to a rapid decrease in IL-17RA in FBXW11 WT cells but not in FBXW11 KO cells (Figure 4C, left panel). It is known that there is functional redundancy between FBXW1A and FBXW11 [54]. FBXW1A siRNA was transiently delivered into both A549 FBXW11 WT and KO cells before treatment with CHX. This showed that knock-down of FBXW1A slightly stabilized endogenous IL-17RA compared to the control siRNA group. To further verify our findings, we used the CRISPR/Cas9 technique to knock out FBXW11 in the Ishikawa cell line. Although the basal level of endogenous IL-17RA protein was dramatically increased in the Ishikawa FBXW11 KO cells compared to the parental WT cells, knock-down of FBXW1A did not show any obvious effects on the protein stability of endogenous IL-17RA (Figure 4C, right panel). These findings suggest that FBXW11 KO increases the protein stability of endogenous IL-17RA, while the effects of FBXW1A knock-down are dependent on the cellular context, as shown in previous reports (comprehensively reviewed in [55]).

### 3.6. Expression Levels of FBXW11 and IL-17RA Are Inversely Correlated

Due to a lack of good antibodies to detect endogenous FBXW11 (of note, we tested five different commercial antibodies, but none of them worked), we used real-time qPCR analysis to measure FBXW11 mRNA levels and Western blot analysis to quantify IL-17RA protein levels in 12 human cell lines (Figure 5A,B). Our results showed an inverse relationship between FBXW11 mRNA levels and IL-17RA protein levels, such that high FBXW11 mRNA levels were associated with low IL-17RA protein levels and vice versa (Figure 5C). Pearson’s correlation analysis showed that FBXW11 mRNA levels and IL-17RA protein levels were significantly inversely correlated (Figure 5D). We further explored a public protein database, the Clinical Proteomic Tumor Analysis Consortium (CPTAC), through the UALCAN platform (https://ualcan.path.uab.edu/analysis-prot.html, accessed on 1 December 2022) to analyze FBXW11 and IL-17RA protein levels across multiple cancer types. Our analysis showed that IL-17RA protein levels were significantly higher in brain tumors and uterine tumors than in the corresponding normal control tissues. On the other hand, FBXW11 protein levels were significantly lower in brain tumors and uterine tumors than in the corresponding normal control tissues (Figure 5E). Pearson’s correlation analysis also revealed a significant inverse correlation between the protein levels of FBXW11 and IL-17RA in both the brain tissues (Figure 5F) and the uterine tissues (Figure 5G). Phosphorylation abundance of IL-17RA was analyzed using the LinkedOmicsKB platform (https://kb.linkedomics.org/, accessed on 1 June 2023) and found that the levels of IL-17RA phosphorylation at S629 and S708 in the uterine tumor samples were significantly lower than in the corresponding normal controls (Appendix A). The levels of IL-17RA phosphorylation at S801 in the uterine tumor samples were also lower than in the corresponding normal controls, but there was no statistically significant difference, likely due to the smaller sample size (Appendix A). These findings are consistent with our hypothesis that phosphorylation accelerates IL-17RA degradation [25].

### 3.7. 665-804 Domain of IL-17RA Determines Its Protein Stability and Ubiquitylation Mediated by FBXW11

A previous report has shown that expression of the IL-17RA ∆665 mutant was more robust than that of the full-length IL-17RA. Furthermore, treatment with TNF-α and IL-17A resulted in higher levels of IL-6 secretion in IL-17RA-/- fibroblasts expressing the IL-17RA ∆665 mutant compared to cells expressing the full-length IL-17RA [10]. We generated two truncation mutants of human IL-17RA, including Flag-IL-17RA ∆665-804 and Flag-IL-17RA ∆729-773 [25]. Not only are the putative candidate phosphodegrons located within the truncated 665-804 domain, but there are also multiple potential phosphorylation sites present within the domain (Figure 6A). Our co-IP assays revealed that deletion of amino acids 665-804 resulted in a remarkably lower binding association between truncated IL-17RA and FBXW1A or FBXW11, compared to the full-length IL-17RA. However, deletion of 729-773 did not affect the physical binding compared to the full-length IL-17RA protein (Figure 6B). Our ubiquitylation assay results also showed that deletion of the 665-804 domain dramatically decreased ubiquitylation of IL-17RA compared to the full-length IL-17RA (Figure 6C). Ectopic overexpression of full-length IL-17RA and IL-17RA ∆665-804 in HEK293T cells revealed that protein levels of full-length IL-17RA dramatically decreased following treatment with CHX, whereas protein levels of IL-17RA ∆665-804 did not exhibit this trend (Figure 6D,E). Furthermore, treatment with MG132 significantly stabilized the full-length IL-17RA protein but not the IL-17RA ∆665-804 protein (Figure 6D,F).

### 3.8. Knock-Out of FBXW11 Suppresses Expression of IL-17-Downstream Genes through Inhibiting Nuclear Entry of NF-κB p65

Next, we investigated the functional effects of FBXW11 KO on IL-17 signaling pathways, including Akt, MAPK, and NF-κB. Our previous results showed that knock-out of FBXW11 led to a dramatic stabilization of IL-17RA protein in A549 cells and Ishikawa cells (Figure 4C). We treated A549 FBXW11 WT, A549 FBXW11 KO, Ishikawa FBXW11 WT, and Ishikawa FBXW11 KO cells with 20 ng/mL rhIL-17A for 10 and 30 min and examined the components of IL-17 signaling pathways using Western blot analysis. We found that rhIL-17A treatment dramatically increased the levels of phosphorylated p38 MAPK (p-p38 MAPK) in FBXW11 KO cells but not in FBXW11 WT cells (Figure 7A and Appendix A). The levels of phosphorylated ERK1/2 (p-ERK1/2) were higher in FBXW11 KO cells than those in FBXW11 WT cells after rhIL-17A treatment (Figure 7B and Appendix A). Treatment with rhIL-17A did not increase levels of phosphorylated Akt (p-AKT) in either A549 or Ishikawa cell lines. Phosphorylated JNK (p-JNK) was obviously induced after 30 min of treatment in Ishikawa cells but not in A549 cells (Figure 7B and Appendix A). The basal levels of phosphorylated IκBα (p-IκBα) in FBXW11 KO cells were higher than those in FBXW11 WT cells, and treatment with rhIL-17A for 10 and 30 min also induced more p-IκBα in FBXW11 KO cells compared to FBXW11 WT cells. Correspondingly, a decrease in IκBα levels was observed in both the FBXW11 WT and KO cell lines, but the levels of IκBα in FBXW11 KO cells were not lower than those in FBXW11 WT cells after rhIL-17A treatment (Figure 7C and Appendix A). Moreover, in our cell fractionation experiments, we observed that the nuclear protein levels of NF-κB p65 in FBXW11 KO cells were obviously lower than those in FBXW11 WT cells, both at the basal levels and after rhIL-17A treatment. However, the cytoplasmic NF-κB p65 levels in both the WT and KO cell lines were almost equal, regardless of rhIL-17A treatment (Figure 7D). These results suggest that knock-out of FBXW11 prevents nuclear entry of NF-κB p65. It is well known that the E3 ligases FBXW1A/FBXW11 mediate ubiquitylation and degradation of p-IκBα [26]. In our A549 FBXW11 KO cells, knock-out of FBXW11 increased the basal levels of IκBα (Figure 7C) as ubiquitylation of IκBα was reduced due to lack of FBXW11. Therefore, IκBα continued to bind to and keep NF-κB p65/p50 in the cytoplasm. Even after rhIL-17A treatment, nuclear entry of NF-κB p65 in A549 FBXW11 KO cells was much less than that in A549 FBXW11 WT cells (Figure 7D), resulting in reduced transcriptional activities of NF-κB. Since NF-κB is the main transcription factor that initiates expression of IL-17-downstream genes, we predicted that IL-17 cytokines might fail to induce expression of IL-17-downstream genes in FBXW11 KO cells. To verify our prediction, we applied real-time qPCR assays to analyze the expression of IL-17-downstream genes. Since IL-17RA is a common subunit that dimerizes with other IL-17Rs to interact with different IL-17 cytokines (IL-17A, IL-17B, IL-17C, IL-17E, and IL-17F), we treated A549 FBXW11 WT and A549 FBXW11 KO cells with 20 ng/mL rhIL-17A, rhIL-17B, rhIL-17C, rhIL-17E, and rhIL-17F for 2 h, individually. Treatment with rhIL-17A significantly increased expression levels of CXCL1 (Figure 7E), CXCL2 (Figure 7F), CXCL8 (Figure 7G), and IL-6 (Figure 7I) in A549 FBXW11 WT cells, while other IL-17 cytokines only significantly increased the levels of IL-6 expression (Figure 7I). As predicted, we observed that knock-out of FBXW11 significantly decreased the expression levels of IL-17-downstream genes upon treatment with IL-17 cytokines (Figure 7E,I). Real-time qPCR analysis of the same IL-17-downstream genes in the Ishikawa FBXW11 WT and FBXW KO cell lines showed similar results (Appendix A).

## 4. Discussion

IL-17-mediated inflammation is critical for innate and adaptive immune responses. Various interaction forms between IL-17 cytokines and receptors have attracted researchers to deepen their understanding of expression patterns, localization, and the roles of each single IL-17 cytokine and receptor under different conditions [1,9,56]. In addition, most of the research activities have been focused on the modulation of downstream signaling pathways of IL-17 receptors and their roles in physiological and pathological situations [57,58,59,60]. PTMs of proteins, such as phosphorylation and ubiquitylation, play significant roles in protein quality control, protein activity, gene expression, inter-/intra-cellular communications, and cellular activities [12,61,62,63]. However, few studies have investigated PTMs of IL-17 receptors. To our best knowledge, our group first reported that IL-17RA was constitutively phosphorylated by GSK3 at threonine 780 (T780), leading to ubiquitylation and degradation, and that IL-17RA phosphorylation was reduced in prostate cancer tissues compared to normal control tissues [25]. However, the specific E3 ligase mediating ubiquitylation of IL-17RA has not been identified. Another group reported that ubiquitylation of IL-17RA by TRAF6 upon IL-17F stimulation is required for downstream signaling [64]. In the current study, we determined that the SCF^FBXW11^ complex mediated ubiquitylation of IL-17RA through recognizing the 665-804 domain of IL-17RA.

We first used MG132 and MLN4924 treatments to verify that degradation of exogenous and endogenous IL-17RA was mediated by the proteasome, particularly by the Cullin-Ring E3 ligase (CRL) complex (Figure 1A–F). In addition, we used an FDA-approved proteasome inhibitor, bortezomib, to further confirm that degradation of endogenous IL-17RA was mediated by the proteasome (Appendix A). A lysosome inhibitor, imidazole, was used to exclude the possibility of lysosome-mediated degradation of IL-17RA (Appendix A).

More than 600 E3 ligases are encoded by the human genome and more than 200 of them belong to the CRL E3 ligase family [16]. To narrow down the numbers of E3 ligase candidates targeting IL-17RA for ubiquitylation, we carried out phosphodegron predictions. A phosphodegron is a short linear motif that can be recognized by E3 ligases after phosphorylation. FBXW7 mediates the ubiquitylation and degradation of multiple oncogenes, such as c-Myc and c-Jun [43,65,66]. In our prediction, a short linear motif (780-TPYEEE-785) of human IL-17RA matched with a phosphodegron, TPxxE, which is recognized by FBXW7 (Appendix A) [67]. On the other hand, another short linear motif (725-DSPLGSST-732) of human IL-17RA also matched with a phosphodegron, DSGxxST, which is recognized by FBXW1A/FBXW11 (Appendix A) [54]. The roles of FBXW1A/FBXW11 are context-dependent, and FBXW1A and FBXW11 are believed to be functionally redundant [68]. When IL-17-binding proteins were precipitated by IL-17A cytokines, E3 ligases FBXW1A/FBXW11 and scaffold protein Cullin1 were among the components of the IL-17-binding proteins [51,69]. In addition, IL-17RA was listed among thousands of candidate substrates that might bind with FBXW11 using a parallel adaptor capture proteomics (PAC) approach (see Appendix A of [70]), but IL-17RA was not ranked high enough to be considered as an FBXW11 substrate by the investigators. Based on our phosphodegron predictions and the published hints, we hypothesized that FBXW7, FBXW1A, and FBXW11 might be the candidate E3 ligases. These three E3 ligases belong to the FBXW family with 10 members that interact with phosphodegrons of a specific substrate through the WD repeat domain [48]. To determine which specific FBXW family member binds to IL-17RA, we performed co-IP assays and showed that FBXW1A, FBXW5, FBXW7, FBXW9, and FBXW11 had the highest binding association towards IL-17RA (Figure 2A,B). Since the WD repeat domain is a conserved domain and percent identity analysis showed that FBXW1A, FBXW7, and FBXW11 bear the greatest sequence similarities (Appendix A), it is reasonable that multiple FBXW family members show various degrees of binding to IL-17RA. Another possible reason could be that our co-IP assays were conducted in an overexpressed system such that non-specific binding could not be completely avoided. Since the functions of SCF E3 ligases are to ubiquitylate the substrates, we performed ubiquitylation assays using two different approaches, and both consistently demonstrated that FBXW11 had the highest E3 ligase activity among the candidates (Figure 3A and Appendix A). Furthermore, we confirmed that FBXW11-mediated ubiquitylation of IL-17RA was dose-dependent (Figure 3B,C), implying that the ubiquitylation activity of FBXW11 towards IL-17RA is specific. Further, the engineered ubiquitin variant Ubv.Fw11.2 slightly decreased basal ubiquitylation levels of endogenous IL-17RA by FBXW11 in THP-1 cells (Figure 3B). Taken together, our findings suggest that FBXW11 is the specific E3 ligase that ubiquitylates IL-17RA.

It has been well documented that K48-linked polyubiquitin participates in proteasomal degradation of proteins and that K63-linked polyubiquitin is involved in the activation of protein and signaling transduction [71,72,73]. We found that single K27R mutation of the ubiquitin remarkably decreased ubiquitylation levels of endogenous IL-17RA in both THP-1 and HCT116 cell lines, although a single mutation of other lysine residues also showed various degrees of reduced ubiquitylation (Figure 3D,E). These findings suggest that K27 is critical for FBXW11-mediated ubiquitylation of IL-17RA. It is possible that other lysine linked-polyubiquitin chains are secondary to K27-linked polyubiquitin. Since mixed ubiquitylation-dependent regulation of protein stability has been documented [72,74,75], we cannot exclude the possibility that FBXW11 ubiquitylates IL-17RA through mixed polyubiquitin chains. Beyond K48- and K63-linked polyubiquitination, polyubiquitination mediated by other lysine residues is termed non-canonic ubiquitylation, and it is poorly understood so far [73]. K27-linked non-canonic ubiquitylation is indispensable in immune responses, cytokine signaling, and T cell activation and differentiation [21]. IL-17RA is a key receptor in the IL-17 signaling and Th17 inflammatory responses; therefore, our discovery further demonstrates the importance of K27-linked polyubiquitylation in the immune system.

We found that overexpression of FBXW11 accelerated degradation of IL-17RA (Figure 4A,B) while knock-out of FBXW11 increased the protein stability of IL-17RA (Figure 4C). Although knock-out of FBXW11 did not completely block the degradation of IL-17RA, which was attributed to FBXW1A compensation in the A549 cell line but not in the Ishikawa cell line [76], we demonstrated an inverse correlation between FBXW11 mRNA levels and IL-17RA protein levels in 12 human cell lines, including immortalized normal cell lines and cancer cell lines (Figure 5A–D). Further exploration of a public proteomics database also showed that FBXW11 protein levels were inversely correlated with IL-17RA protein levels in the brain and uterine tissues (Figure 5E–G). These results suggest that FBXW11 down-regulates IL-17RA protein levels in both human cell lines and human tissues.

The intracellular domain of IL-17RA beyond the SEFIR domain has been shown to be critical for IL-6 secretion after stimulation with IL-17 and TNFα, and the protein levels of the IL-17RA ∆665 truncation mutant were robustly higher than those of the full-length IL-17RA [10]. We generated two truncation mutants (IL-17RA ∆729-773 and IL-17RA ∆665-804) and demonstrated that deletion of ∆665-804 not only reduced binding with FBXW1A and FBXW11, but also decreased ubiquitylation mediated by FBXW11 (Figure 6A–C). Further, the IL-17RA ∆665-804 truncation mutant was more stable than the full-length IL-17RA (Figure 6D–F). These findings indicate that the 665–804 domain is critical for ubiquitylation and degradation of IL-17RA. However, we did not examine which amino acid residue of IL-17RA is responsible for phosphorylation and subsequent ubiquitylation in the present study, which remains to be determined in future studies. Ubiquitylation is largely dependent on the priming phosphorylation of the substrates [43,77], and we predicted a candidate phosphodegron recognized by FBXW1A and FBXW11 together with multiple potential phosphorylation sites in the 665-804 domain. We believe that phosphorylation-dependent ubiquitylation of IL-17RA happens within this domain. The levels of IL-17RA phosphorylation at S708 and S629 in the uterine tumor samples were significantly lower than in the corresponding normal controls (Appendix A), while the levels of IL-17RA protein were higher in the uterine tumors than in the normal tissues, suggesting that reduced IL-17RA phosphorylation is linked to more stable IL-17RA protein due to less ubiquitylation and degradation. Further study is needed to illustrate the specific amino acids of IL-17RA that are involved in phosphorylation and ubiquitylation, which could be explored as therapeutic targets using the proteolysis-targeting chimera (PROTAC) technique in the treatment of autoimmunity and cancer [78].

Since knock-out of FBXW11 remarkably increased endogenous IL-17RA protein levels in the A549 and Ishikawa cell lines, we hypothesized that IL-17 cytokines that interact with IL-17RA should activate IL-17 signaling pathways, inducing higher levels of expression of downstream genes in FBXW11 KO cells compared to the parental FBXW11 WT cells. However, we found that nuclear levels of NF-κB p65 in FBXW11 KO cells were much less than in FBXW11 WT cells after 30 min of treatment with rhIL-17A (Figure 7D). This raised a question why nuclear entry of NF-κB p65 was decreased in FBXW11 KO cells. Normally, IL-17 signaling is initiated by IL-17A binding to the receptors, leading to phosphorylation of IκBα. Phosphorylated IκBα is ubiquitylated by FBXW1A/FBXW11, resulting in proteasome-mediated degradation [26]. Degradation of IκBα liberates NF-κB p50/p65 to enter the nucleus, thus starting transcription of downstream genes [1,79]. Yet FBXW11 knock-out led to a failure in the ubiquitylation and degradation of IκBα, allowing IκBα to continue to trap NF-κB p50/p65 in the cytoplasm and subsequently fail to initiate expression of downstream genes. Furthermore, our real-time qPCR analysis confirmed the lower levels of basal and induced expression of downstream genes in FBXW11 KO cells compared to FBXW11WT cells (Figure 7E–I and Appendix A). In addition, as a core component of IL-17 signaling, Act1 has been shown to be ubiquitylated by FBXW1A/FBXW11 and degraded by the proteasome [23]. Overall, IL-17 signaling is finely regulated by FBXW1A/FBXW11 at different levels, including IL-17RA, Act1, and IκBα (Figure 8). The present study found that IL-17RA was ubiquitylated by FBXW11, followed by proteasomal degradation. Knock-out of FBXW11 stabilizes IL-17RA and Act1, which is supposed to enhance IL-17 signaling and induce more expression of downstream genes. However, knock-out of FBXW11 decreases degradation of IκBα. Therefore, the end readouts of IL-17 signaling, that is, the expression levels of downstream genes, are reduced in FBXW11 KO cells. Treatment with rIL-17A induced more activation of MAPK signaling pathways in FBXW11 KO cells than in WT cells, but the MAPK signaling pathways might not be involved in the expression of the IL-17-downstream genes examined. The biological significance of MAPK activation in this setting remains unknown, which should be investigated in future studies.

## 5. Conclusions

In summary, the present study identified SCF^FBXW11^ as a critical E3 ligase that regulates the IL-17 signaling pathway at the IL-17RA level. Future studies may be conducted to explore the potential of targeting SCF^FBXW11^ for the treatment of IL-17-dependent inflammatory and autoimmune conditions as well as cancers.

## Figures and Tables

**Figure 1 biomedicines-12-00755-f001:**
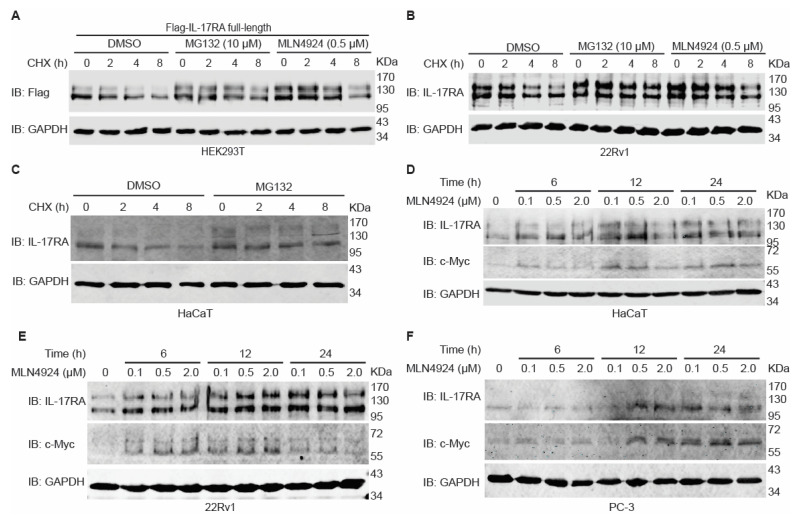
Exogenous and endogenous IL-17RA are degraded through the ubiquitin–proteasome system, particularly the Cullin-Ring-E3 ligase complex. (**A**) Western Blot analysis of expression levels of exogenous Flag-IL-17RA. A quantity of 5 μg full-length Flag-IL-17RA plasmids was transiently transfected into HEK293T cells in a 10 cm dish. Eighteen hours (h) post-transfection, the cells were evenly split into 6-well plates. Then, 42 h post-transfection, the cells were treated with 20 μM MG132 or 0.5 μM MLN4924 for 8 h and 50 μg/mL CHX for indicated hours. Of note, in the group with CHX treatment for 8 h, MG132 or MLN4924 was added to all treatment groups simultaneously. Treatment of each subgroup was terminated at the same time; therefore, CHX was added to another two subgroups 2 and 4 h before collecting cells. Overall, the treatment time of MG132 and MLN4924 was 8 h, and the treatment times of CHX were 0, 2, 4, and 8 h. DMSO was used as the control treatment. (**B**) Western blot analysis of endogenous IL-17RA in 22Rv1 cells after treatment with 20 μM MG132 and 0.5 μM MLN4924 for 8 h and 50 μg/mL CHX for indicated hours. DMSO was used as the control treatment. (**C**) Western blot analysis of endogenous IL-17RA in the HaCaT cell line after treatment with 10 μM of MG132 for 8 h and 50 μg/mL of CHX for indicated times. DMSO was used as the control treatment. (**D**–**F**) The neddylation inhibitor MLN4924 at 0.1 μM, 0.5 μM, and 2.0 μM was used to treat HaCaT (**D**), 22Rv1 (**E**), and PC-3 (**F**) cell lines for 6, 12, and 24 h. DMSO was used as the control treatment. Each experiment was repeated at least three times.

**Figure 2 biomedicines-12-00755-f002:**
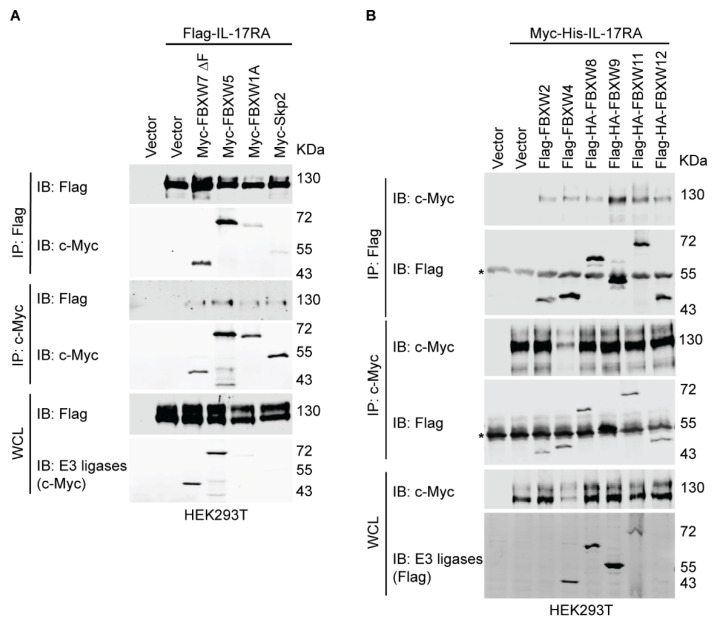
Several FBXW E3 ligases bind to IL-17RA. (**A**) Binding of IL-17RA with E3 ligases. HEK293T cells were seeded at a density of 1 × 10^6^ into 6 cm dishes. At 20 h post-seeding, 1.5 µg full-length Flag-IL-17RA, 1.5 µg Myc-FBXW7 ∆F, 1.5 µg Myc-FBXW5, 1.5 µg Myc-FBXW1A, and 1.5 µg Myc-Skp2 plasmids were transiently transfected using jetPRIME transfection reagent as indicated. An empty vector was used to compensate for the total amount of plasmids. Then, 48 h post-transfection, proteins were extracted using IP lysis buffer. The reciprocal co-IP assays were carried out using 1 µg anti-Flag M2 or 1 µg anti-c-Myc antibodies. The experiments were repeated four times independently. (**B**) Binding of IL-17RA with E3 ligases. HEK293T cells were seeded at a density of 1 × 10^6^ into 6 cm dishes. Then, 2 µg full-length Myc-His-IL-17RA, 2 µg Flag-FBXW2, 2 µg Flag-FBXW4, 1.5 µg Flag-HA-FBXW8, 1 µg Flag-HA-FBXW9, 2 µg Flag-HA-FBXW11, and 2 µg Flag-FBXW12 plasmids were transiently transfected using jetPRIME transfection reagent as indicated. An empty vector was used to compensate for the total amount of plasmids. At 48 h post-transfection, proteins were extracted using IP lysis buffer. The reciprocal co-IP assays were carried out using 1 µg anti-Flag M2 or 1 µg anti-c-Myc antibodies. The experiments were repeated three times independently. Asterisks indicate the IgG heavy chain of the antibodies used in the co-IP assays.

**Figure 3 biomedicines-12-00755-f003:**
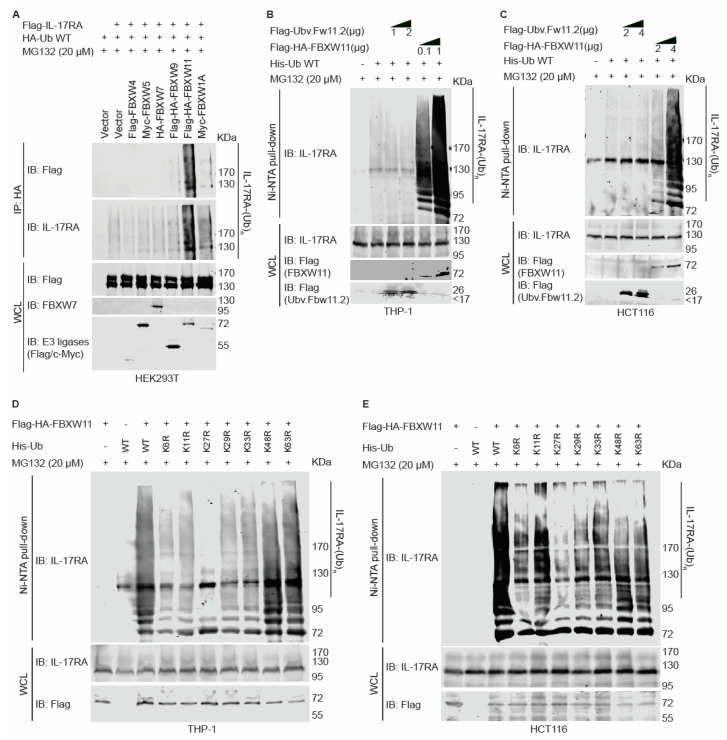
FBXW11 ubiquitylates IL-17RA mainly via K27-linked polyubiquitin chains in a dose-dependent way. (**A**) HEK293T cells were seeded into 10 cm dishes at a density of 4 × 10^6^. Then, 3 µg full-length Flag-IL-17RA, 2 µg HA-ubiquitin WT, 3 µg Flag-FBXW4, 3 µg Myc-FBXW5, 4 µg HA-FBXW7, 2 µg Flag-HA-FBXW9, 3 µg Flag-HA-FBXW11, and 3 µg Myc-FBXW1A plasmids were transiently transfected using jetPRIME transfection reagent as indicated. An empty vector was used to compensate for the total amount of plasmids. At 42 h post-transfection, transfected cells were treated with 10 µM MG132 for 6 h. The experiments were repeated four times independently. (**B**) THP-1 cells were seeded into 10 cm dishes at a density of 1.5 × 10^6^. At 24 h post-seeding, 1 or 2 µg Flag-Ubv.Fw.11.2, 0.1 or 1 µg Flag-HA-FBXW11, and 2.5 µg His-ubiquitin WT plasmids were transfected using JetPrime transfection reagent. An empty vector was used to compensate for the total amount of plasmids. At 42 h post-transfection, 20 µM MG132 was added to treat the cells for 6 h. (**C**) HCT116 cells were seeded into 10 cm dishes at a density of 2 × 10^6^. At 24 h post-seeding, 2 or 4 µg Flag-Ubv.Fw.11.2, 2 or 4 µg Flag-HA-FBXW11, and 2.5 µg His-ubiquitin WT plasmids were transfected using JetPrime transfection reagent. An empty vector was used to compensate for the total amount of plasmids. At 42 h post-transfection, 20 µM MG132 was added to treat the cells for 6 h. The experiments were repeated four times independently. (**D**) THP-1 cells were seeded into 10 cm dishes at a density of 1.5 × 10^6^. At 24 h post-seeding, 1 µg Flag-HA-FBXW11 and 2.5 µg WT or single lysine mutated His-ubiquitin plasmids were transfected using JetPrime transfection reagent. An empty vector was used to compensate for the total amount of plasmids in transfection. At 42 h post-transfection, 20 µM MG132 was added to treat the cells for 6 h. The experiments were repeated four times independently. (**E**) HCT116 cells were seeded into 10 cm dishes at a density of 3 × 10^6^. At 24 h post-seeding, 2 or 4 µg Flag-HA-FBXW11 and 2.5 µg WT or single lysine mutated His-ubiquitin plasmids were transfected using JetPrime transfection reagent. An empty vector was used to compensate for the total amount of plasmids in transfection. At 42 h post-transfection, 20 µM MG132 was added to treat the cells for 6 h. The experiments were repeated three times independently.

**Figure 4 biomedicines-12-00755-f004:**
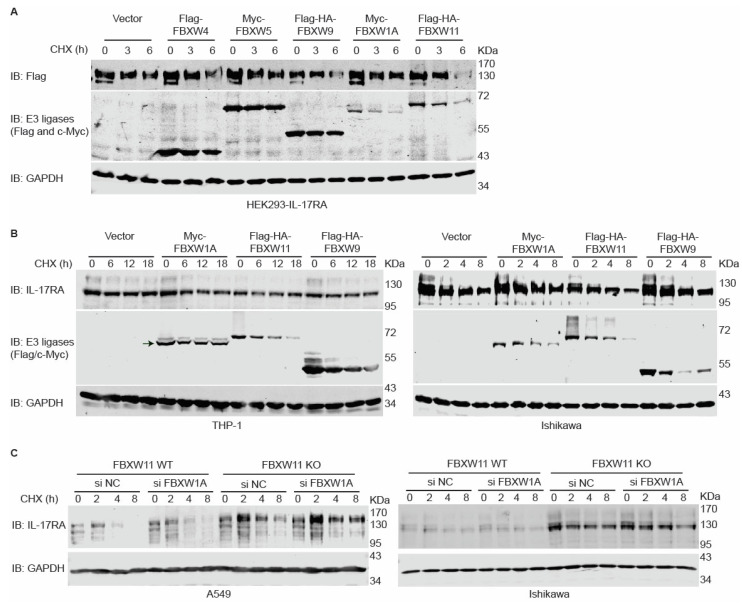
FBXW11 down-regulates the protein level of IL-17RA. (**A**) CHX assays were conducted to investigate the effects of FBXW1A, FBXW9, and FBXW11 overexpression on an HEK293-IL-17RA stable cell line. Twenty-four hours before transfection, 1.2 × 10^6^ HEK293-IL-17RA cells were seeded into 6 cm dishes. Then, 2 μg Flag-FBXW4, 1.5 μg Myc-FBXW5, 1.5 μg Flag-HA-FBXW9, 3 μg Myc-FBXW1A, or 3 μg Flag-HA-FBXW11 plasmids were transiently transfected using jetPRIME transfection reagent. An empty vector was used as a transfection control and to compensate for the total amount of plasmids for each transfection. Six hours post-transfection, the cells in each 6 cm dish were evenly split into three wells of a 6-well plate. Twenty-four hours post-transfection, the cells were treated with 100 μg/mL CHX for 0, 3, and 6 h, respectively. DMSO was used as the control treatment. (**B**) CHX assays were conducted to investigate the effects of FBXW1A, FBXW9, and FBXW11 on endogenous IL-17RA protein levels in the THP-1 (**left panel**) and Ishikawa (**right panel**) cell lines. For the THP-1 cells, 1.0 × 10^6^ cells were seeded into 6 cm dishes 24 h before transfection, and 4 μg Myc-FBXW1A, 1.5 μg Flag-HA-FBXW9, or 5 μg Flag-HA-FBXW11 was transiently transfected using jetPRIME transfection reagent. An empty vector was used as a transfection control and to compensate for the total amount of plasmids in each transfection. Twenty-four hours post-transfection, the cells were treated with 100 μg/mL CHX for 0, 6, 12, and 18 h, respectively, with DMSO used as the control treatment. For the Ishikawa cells, 3.5 × 10^6^ cells were seeded into 10 cm dishes 24 h before transfection, and 9 μg Myc-FBXW1A, 3 μg Flag-HA-FBXW9, or 10 μg Flag-HA-FBXW11 plasmids was transiently transfected using jetPRIME transfection reagent. An empty vector was used as a transfection control and to compensate for the total amount of plasmids in each transfection. Twelve hours post-transfection, the cells in each 10 cm dish were evenly split into four 6 cm dishes. Thirty-six hours post-transfection, the cells were treated with 100 μg/mL CHX for 0, 2, 4, and 8 h, respectively, with DMSO used as the control treatment. The experiments were independently repeated three times. The arrow indicates the band of Myc-FBXW1A. (**C**) CHX assays were conducted to investigate the effects of FBXW11 knock-out combined with/without FBXW1A knock-down on A549 (**left panel**) and Ishikawa (**right panel**) cell lines. For both cell lines, 2.0 × 10^6^ FBXW11 WT or 2.5 × 10^6^ FBXW11 KO cells were seeded into 10 cm dishes 24 h before transfection, and 20 nM FBXW1A siRNAs were transiently transfected using jetPRIME transfection reagent. Negative control siRNAs were used as a control. Twenty-four hours post-transfection, the cells in each 10 cm dish were split evenly into four 6 cm dishes. Forty-eight hours post-transfection, the cells were treated with 50 μg/mL CHX for 0, 2, 4, and 8 h, respectively, with DMSO used as a control treatment. The experiments were independently repeated three times.

**Figure 5 biomedicines-12-00755-f005:**
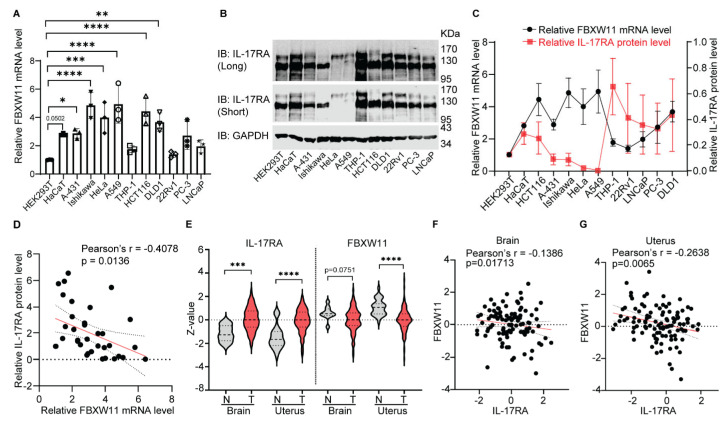
FBXW11 and IL-17RA are inversely correlated. (**A**) Total RNA was isolated from 12 cell lines, including HEK293T, HaCaT, A-431, Ishikawa, HeLa, A549, THP-1, HCT116, DLD1, 22RV1, PC-3, and LNCaP. Real-time qPCR analysis was conducted to measure FBXW11 mRNA levels. The HEK293T cell line was used as a calibration control, and the Student’s *t*-test was used to determine statistical significance when compared to HEK293T. * *p* < 0.05,** *p* < 0.01, *** *p* < 0.001, and **** *p* < 0.0001. Data were collected from three independent biological replicates. (**B**) Western blot analysis was used to determine IL-17RA protein levels across 12 cell lines, including HEK293T, HaCaT, A-431, Ishikawa, HeLa, A549, THP-1, HCT116, DLD1, 22RV1, PC-3, and LNCaP. “Long” indicates long exposure and “Short” indicates short exposure. Signal intensities of IL-17RA and Glyceraldehyde-3-Phosphate Dehydrogenase (GAPDH) in each cell line were determined using Image Studio (Lite Ver 5.2, Li-Cor) software. The ratio of IL-17RA/GAPDH was then calculated to compare relative protein levels. The HEK293T cell line was used as a calibration control. Data were collected from three independent replicates. (**C**) The quantification results shown in (**A**,**B**) were plotted to illustrate the trajectory of relative FBXW11 mRNA levels and IL-17RA protein levels across the 12 human cell lines. Error bars represent means ± standard deviations (SDs). (**D**) Pearson’s correlation analysis was conducted using the data of relative FBXW11 mRNA levels and relative IL-17RA protein levels. (**E**) Proteomic data obtained from the CPTAC database demonstrated IL-17RA and FBXW11 protein levels among glioblastoma multiform (brain tumor), uterus corpus endometrial cancer (UCEC, uterine tumor), and corresponding normal control tissues. Z-values indicate standard deviations from the medians across the samples for the given cancer type. The Student’s *t*-test was used to determine statistical significance between normal control tissues and tumors. *** *p* < 0.001 and **** *p* < 0.0001. (**F**) Pearson’s correlation analysis was conducted using the proteomic data on FBXW11 and IL-17RA protein levels of glioblastoma multiform and normal control tissues. (**G**) Pearson’s correlation analysis was conducted using the proteomic data on FBXW11 and IL-17RA protein levels of UCEC and normal control tissues. Control tissues of glioblastoma were from the frontal cortex. Control tissues of uterine tumors were from endometrium (with or without enrichment) and myometrium.

**Figure 6 biomedicines-12-00755-f006:**
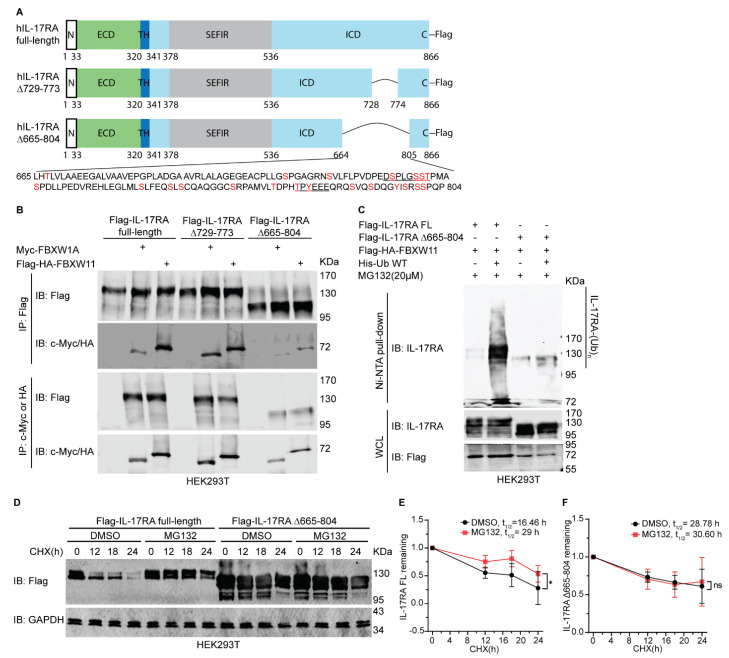
The 665-804 domain of IL-17RA is required for ubiquitylation and protein stability of IL-17RA. (**A**) Schematic diagram of full-length IL-17RA (Flag-IL-17RA FL) and the truncated mutants (Flag-IL-17RA ∆729-773 and Flag-IL-17RA ∆665-804) used in this study. (**B**) Binding of full-length IL-17RA and truncated mutants with FBXW1A and FBXW11. HEK293T cells were seeded into 10 cm dishes at a density of 2.0 × 10^6^. At 20 h post-seeding, 1.5 μg Flag-IL-17RA FL, 1.5 μg Flag-IL-17RA ∆729-773, 1 μg Flag-IL-17RA ∆665-804, 1.5 μg Myc-FBXW1A, and 1.5 μg Flag-HA-Fbxw11 were transiently transfected using jetPRIME transfection reagent as indicated. An empty vector was used to compensate for the total amount of plasmids in transfection. Forty-eight hours post-transfection, the whole cell lysates were extracted for subsequent co-IP and Western blot analyses. (**C**) Ubiquitylation of Flag-IL-17RA ∆665-804 by FBXW11 was less than that of Flag-IL-17RA FL. HEK293T cells were seeded into 10 cm dishes at a density of 4.5 × 10^6^. At 24 h post-seeding, 1 µg Flag-IL-17RA full-length, 0.75 µg Flag-IL-17RA ∆665-804, 3.5 µg Flag-HA-FBXW11, and 2.5 µg His-ubiquitin WT plasmids were transfected using jetPRIME transfection reagent as indicated. Forty hours post-transfection, the cells were treated with 20 µM MG132 for 8 h. Precipitates pulled down by Ni-NTA resins, and corresponding whole cell lysates (WCLs) were subjected to Western blot analysis. (**D**) Western blot analysis of protein stability of Flag-IL-17RA FL and Flag-IL-17RA ∆665-804. Quantities of 1.5 µg Flag-IL-17RA FL or 1.5 µg Flag-IL-17RA ∆665-804 plasmids were transiently transfected into 1.5 × 10^6^ HEK293T cells in a 6 cm dish. Twenty-four hours post-transfection, the cells were treated with 10 μM MG132 for 24 h and 50 μg/mL CHX for the indicated times. DMSO was applied as a control treatment. The experiments were repeated four times independently. (**E**) Quantification of the ratio of exogenous Flag-IL-17RA FL/GAPDH after treatment with MG132 and CHX. “t1/2” means half-life of IL-17RA FL. Statistical significance was conducted using a two-way ANOVA with Šídák’s multiple comparison test. Error bars represent means ± standard deviations (SDs). * *p* < 0.05. (**F**) Quantification of ratio of exogenous Flag-IL-17RA ∆665-804 to GAPDH after treatment with MG132 and CHX. “t1/2” means half-life of IL-17RA ∆665-804. Statistical significance was computed using a two-way ANOVA with Šídák’s multiple comparison test. Error bars represent means ± standard deviations (SDs). “ns” means no statistical significance. The experiments were repeated three times independently.

**Figure 7 biomedicines-12-00755-f007:**
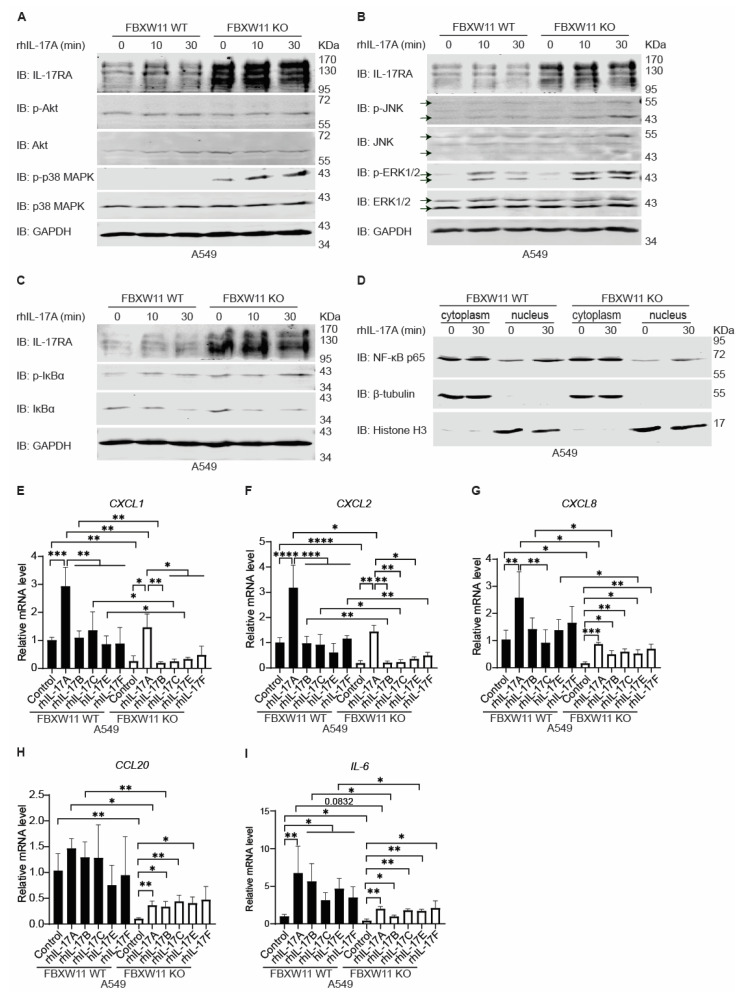
Knock-out of FBXW11 suppresses expression of IL-17-downstream genes through inhibiting nuclear entry of NF-κB p65. (**A**–**C**) Western blot analysis of IL-17RA, phosphorylated AKT (p-AKT), AKT, phosphorylated p38 MAPK (p-p38 MAPK), p38 MAPK, phosphorylated JNK (p-JNK), JNK, phosphorylated ERK1/2 (p-ERK1/2), phosphorylated IκBα (p-IκBα), and IκBα. Quantities of 2 × 10^6^ A549 FBXW11 WT cells and 2.5 × 10^6^ A549 FBXW11 KO cells were treated with 20 ng/mL rhIL-17A for 10 min and 30 min, while the control cells were treated with 0.1% BSA. The arrows indicate the bands of p-JNK, JNK, p-ERK1/2, and ERK1/2. (**D**) Western blot analysis of NF-κB p65 in cytoplasmic and nuclear extracts. Quantities of 2 × 10^6^ A549 FBXW11 WT cells and 2.5 × 10^6^ A549 FBXW11 KO cells were treated with 20 ng/mL rhIL-17A for 30 min, while the control cells were treated with 0.1% BSA. The experiments were repeated six times independently. (**E**–**I**) Induction of IL-17-downstream gene expression. A549 FBXW11 WT and FBXW11 KO cells were treated with 20 ng/mL recombinant human IL-17 (rhIL-17) cytokines, including rhIL-17A, rhIL-17B, rhIL-17C, rhIL-17E, and rhIL-17F, for 2 h. Expression of *CXCL1* (**E**), *CXCL2* (**F**), *CXCL8* (**G**), *CCL20* (**H**), and *IL-6* (**I**) was evaluated using real-time qPCR analysis, normalized to internal GAPDH control. The cells treated with 0.1% bovine serum albumin (BSA) were used as calibration controls. The fold change of each target gene over the control is shown. Error bars represent means ± standard deviations (SDs). The Student’s *t*-test was used to calculate the statistical significance of fold changes. * *p* < 0.05, ** *p* < 0.01, *** *p* < 0.001, and **** *p* < 0.0001. The experiments were repeated three times independently.

**Figure 8 biomedicines-12-00755-f008:**
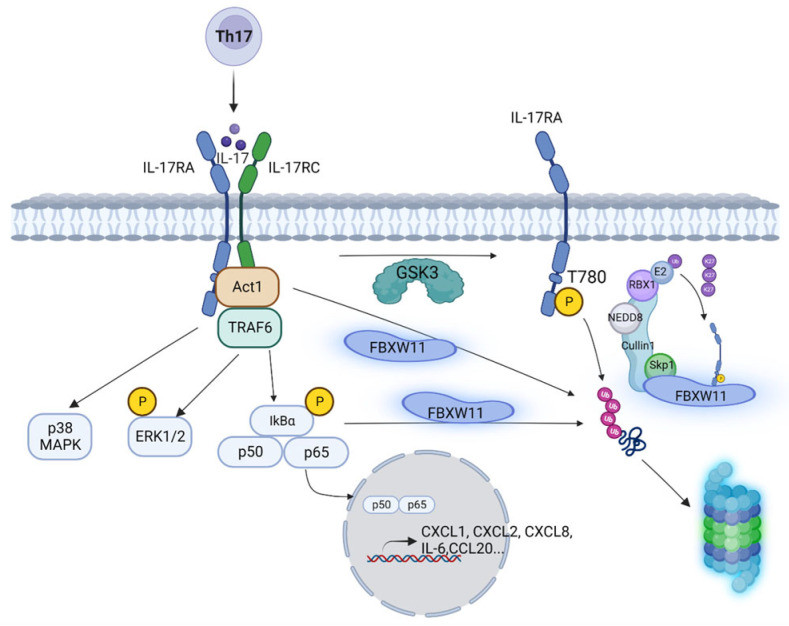
SCF^FBXW11^ complex regulates IL-17 signaling at multiple levels, including IL-17RA, Act1, and IκBα. IL-17A, interleukin-17A; IL-17RA, interleukin-17 receptor A; IL-17RC, interleukin-17 receptor C; Act1, NF-κB-activated protein 1; TRAF6, tumor-necrosis factor receptor-associated factor 6; Ub, ubiquitin; IκBα, NF-κB inhibitor α; p50 and p65, NF-κB subunits; p38 MAPK, p38 mitogen-activated protein kinase; ERK1/2, extracellular signal-regulated kinase 1/2; IL-6, interleukin-6; CXCL1, C-X-C motif ligand 1; CXCL2, C-X-C motif ligand 2; CXCL8, C-X-C motif ligand 8; CCL20, C-C motif ligand 20; GSK3, Glycogen Synthase Kinase; E2, ubiquitin conjugation enzyme E2; RBX1, Ring-Box 1; NEDD8, Neural Precursor Cell Expressed, Developmentally Down-Regulated 8; Skp1, S-Phase Kinase Associated Protein 1; FBXW11, F-Box and WD Repeat Domain Containing 11. Illustration was made using BioRendor.

**Table 1 biomedicines-12-00755-t001:** Primers used in real-time qPCR.

Gene Name	Primers (Synthesized by Eurofins Genomics)
*hCXCL1*	Forward: 5′-AACCGAAGTCATAGCCACAC-3′ Reverse: 5-GTTGGATTTGTCACTGTTCAGC-3′
*hCXCL2*	Forward: 5′-CTGCGCTGCCAGTGCTT-3′ Reverse: 5′-CCTTCACACTTTGGATGTTCTTGA-3′
*hCXCL8*	Forward: 5′-GTGCAGTTTTGCCAAGGAGT-3′ Reverse: 5′-CTCTGCACCCAGTTTTCCTT-3′
*hCCL20*	Forward: 5′-TGCTGTACCAAGAGTTTGCTC-3′ Reverse: 5′-CGCACACAGACAACTTTTTCTTT-3′
*hIL-6*	Forward: 5′-GGTACATCCTCGACGGCATCT-3′ Reverse: 5′-GTGCCTCTTTGCTGCTTTCAC-3′
*hFBXW11*	Forward: 5′-GTGGGATGTGAACACGGGTGA-3′ Reverse: 5′-CGTAAAGTGATGTCGGTCGCAG-3′
*hGAPDH*	Forward: 5′-CCATGGGGAAGGTGAAGGTC-3′ Reverse: 5′-AGTGATGGCATGGACTGTGG-3′

## Data Availability

Please refer to the Materials and Methods section.

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
