# Peer review of "SCFFBXW11 Complex Targets Interleukin-17 Receptor A for Ubiquitin–Proteasome-Mediated Degradation"

_biomedicines, 2024, doi:10.3390/biomedicines12040755_

Round 1
Reviewer 1 Report
Comments and Suggestions for Authors
The authors previously reported that IL-17RA can be polyubiquitinated, leading to proteasomal degradation. However, they did not determine specific E3 ligase responsible for polyubiquitinating IL-17RA. In this manuscript entitled “SCFFBXW11 complex targets interleukin-17 receptor A for ubiquitin-proteasome-mediated degradation” by Jin, B. et al, the authors identified FBXW11 as a ubiquitin E3 ligase promoting polyubiquitination and proteasomal degradation of IL-17RA through Lys27-linked polyubiquitin chain. Considering together with previous reports, the authors further concluded that FBXW11 regulates IL-17 signaling pathway through ubiquitin-dependent proteasome degradation at IL-17RA, Act1 and IkBa levels. The results are clear and convincing. The manuscript is very well written, and the authors carefully discussed about what they can or cannot conclude from each data in the discussion section. However, some experiments in the manuscript still have problem. The following issues should be addressed before this paper can be accepted. I hope these comments will improve the paper. The details are as follows.
(1) In Figure supplement 2A-2D, the authors evaluated the effect of Bortezomib, a proteasome inhibitor, on endogenous IL-17RA protein level in several cell lines. Figure supplement 2A clearly shows that Bortezomib treatment increased IL-17RA protein level in HaCaT cells. However, in Figure supplement 2B-2D, IL-17RA protein level does not seem to be increasing by the treatment of Bortezomib in other cell lines. This is probably due to the weaker activity of Bortezomib to inhibit proteasomal degradation compared to MG132, as the authors described. But, if the authors really want to show these subtle effects of Bortezomib on IL-17RA in these cell lines, they should do the densitometric analysis to measure the intensity of each band and calculate the IL-17RA expression relative to the GAPDH expression and statistically analyze the increase of IL-17RA expression with Bortezomib treatment.
(2) In Figure supplement 2G and 2H, the authors used imidazole as an inhibitor for lysosome to examine the involvement of lysosome in IL-17RA degradation. However, imidazole is not a commonly used inhibitor for lysosome. The authors should therefore do the experiment with more general lysosomal inhibitor, such as ammonia chloride, chloroquine or Bafilomycin A1 to precisely rule out the involvement of lysosome in IL-17RA degradation.
(3) In Figure 7, the authors showed the increase of IL-17RA protein level, p38 MAPK phosphorylation, ERRK1/2 phosphorylation and IkBa phosphorylation, and the decrease of IL-17A-mediated nuclear entry of NF-kB p65 and following expression of CXCL1, 2 and 8 in FBXW11-KO cells. These findings are all compatible with the impaired IL-17RA degradation plus impaired degradation of IkBa. However, IL-17A-mediated IkBa degradation was normally occurred in FBXW11-KO cells. Although the authors emphasized that basal level of IkBa was increased by FBXW11-deficiency, this finding does not match to the impaired IL-17A-mediated nuclear translocation of NF-kB p65, since IkBa degradation and p65 nuclear translocation should be tightly correlated. The authors should therefore explain about the possible reasons for this discrepancy.
Reviewer 2 Report
Comments and Suggestions for Authors
In this manuscript, the authors found that human IL-17RA is targeted by FBXW11 for ubiquitination, and they further revealed that the domain 665-804 of IL-17RA is critical for interaction with FBXW11 and subsequent ubiquitination.
1. The authors performed many immunoblots, but some images are over manipulated. For example, Fig 3E is apparently overexposed, and the same is true for the right panel of Fig 4B for IL-17RA. Please minimally process the figures which should faithfully represent the original images or use other repeated images instead.
2. Please use arrows to indicate each protein in immunoblots since there are unspecific bands for some proteins.
3. Fig 7D shows the wrong size of p65 (the molecular weight of p65 is 65 KDa). In addition, the authors include both p50 and p65 in Fig 8, but they did not detect the protein levels of p50 in this study.
4. In terms of the results shown in Fig 7, only one cell line (A549) was detected. Did the authors find the same or similar results in other cell lines? At least one additional cell line should be examined if the conclusions led by A549 are appropriate.
5. All the gene names should be italicized.
Round 2
Reviewer 1 Report
Comments and Suggestions for Authors
The authors responded to all the comments. Their answers all make sense, so that the manuscript now can be accepted for the paper.
Reviewer 2 Report
Comments and Suggestions for Authors
The authors have adequately addressed all my concerns.